# Glass Fracture during Micro-Scratching

**Islam Zakiev [1], George A. Gogotsi [2], Michael Storchak [3],\* and Vadim Zakiev [1]**

[1] Department of Aircraft Design, National Aviation University, pr. Kosmonavta Komarova 1, 03680 Kiev, Ukraine; zakiev20@ukr.net (I.Z.); zakiev@ukr.net (V.Z.)

[2] G.S.Pisarenko Institute for Problems of Strength National Academy of Sciences of Ukraine, Timiryazevskaya Str. 2, 01014 Kiev, Ukraine; ggogotsi@ipp.kiev.ua

[3] Institute for Machine Tools, University of Stuttgart, Holzgartenstr. 17, 70174 Stuttgart, Germany

\* Correspondence: michael.storchak@ifw.uni-stuttgart.de; Tel.: +49-711-685-83831

**Abstract:** The regularity of glass surface fracture and resistance to destruction were investigated by the methods of progressive and static microscratching with the Berkovich indenter. The research hardware was the original nanoindentation/microscratching devices and a non-contact interference profilometer for studying the morphology of the formed microscratches. The regularities of the fracture stages and the cracks growth along the microscratch were established depending on the indenter applied load. Based on analysis of the microcracks profile formed at various loads on the indenter immediately after the process of applying these scratches and after several hours of rest, it was found that the process of crack propagation along the scratch continues for a long time. Taking into account this established fact, a discrete-statistical method of the cracks formation for a long time is proposed. In accordance with this method, scratching is carried out with a constant load on short and separated tracks. The load on the indenter in each track increases discretely with a certain step. The influence of the medium on the scratching process is analyzed. The breaking mechanism in the glasses scratching process is formulated as the load on the indenter increases, and a model of the glass fracture stages is proposed.

**Keywords:** glass; nanoindentation; microscratching; morphology; brittle fracture

## 1. Introduction

The study of the fracture process of brittle materials' surface layers, in particular various glasses, is of fundamental interest for developing an assessment of their scratch resistance. Glasses are widely used in terrestrial and space conditions as structural materials and the scope of their practical application is expanding more and more, although they are characterized by brittle fracture and low strength [1,2]. The rapid development of information technologies is inextricably connected with the use of fiber optic communication lines, the most important element of which is highly transparent quartz glass. The strength and abrasion resistance of the optical connectors' polished tips (ferrule) largely depends on the bandwidth of the transmitted data. In addition, glass is used to create promising discs for recording and storing information, where high microhardness, mechanical strength and abrasion resistance are the most important requirements. The widespread use of glass displays in modern mobile communications determines the relevance of their scratch resistance [3–5]. Safety glasses of solar panels are also exposed to sand and dust, which adversely affect the efficiency of solar panels, because microcracks prevent light from entering. Over the past several decades, the mechanical properties of glasses have been largely improved to find application in different areas [6]. Therefore, the study of the fracture mechanism and the causes of crack propagation in brittle materials is of significant interest for a large number of different scientific fields and the practical application of such materials [7].

This article explores the features of fracture and resistance of the glass surface to fracture, investigated by the methods of progressive and static microscratching with the Berkovich indenter. The regularities of the fracture stages and the cracks development along the scratch are established depending on the indenter-applied load. A discrete-statistical method is proposed for studying the glass damage stages during scratching as the load increases.

## 2. Glass Breaking Research Methods

There is great need for numerical methods capable of predicting the fracture behavior of glass [8]. G. Gogotsi studied the fracture toughness of ceramics and glasses using the single edge V-notched beam (SEVNB) test, which is based on the breakage of a bent rectangular sample with a V-shaped notch [9]. Later, to study the resistance of brittle materials to the formation and propagation of fracture cracks the edge, a flaking (EF) method was proposed, the essence of which is to determine fracture toughness by flacking the rectangular specimen edge with a standard diamond indenter [10]. During such a test, the surface layer of investigated material is firstly destroyed by a sharp indenter, and only then does a crack propagate in the bulk material. Surface fracture resistance does not consider the fracture toughness of the material, which is determined as the ratio of the fracture load to the fracture distance from the specimen edge [11].

The scratch test is well established technique for material surface layer mechanical properties investigation [12,13], as well as to study the wear of glasses because of its negative impact on service life and product performance [14]. This technique is successfully used to evaluate the condition of a material's subsurface layers by means of establishing the material's resistance to scratching [15]. By using this method, it is possible to map the microhardness of the examined material and determine the tribological properties of subsurface layers [16–18]. This technique also allows one to study the micromechanical properties of glasses, crack resistance, fracture toughness, fatigue and abrasion resistance at the micro and nano scales under experimental conditions that are closest to the actual operational or technological processes [19].

It should be noted that there is some difference between indentation and scratch hardness in glass [20]. S. Sawamura and L. Wondraczek studied scratch and indentation hardness of a broad variety of glassy materials and found that there is no simple relation between both values [21]. This should be taken into account when developing new materials with improved mechanical properties [22,23].

It is well known that fracture properties of glass are affected by the environment [24]. The results of experimental studies of glasses by sclerometry in various media show that the presence of water in the contact changes the damage mechanisms and the need for further research in this direction was noted [25,26]. Studies of scratching the glass surface and adjacent boundary layers have established differences in fracture resistance depending on the humidity and glass composition [25]. W. Gu and Z. Yao studied the types of surface crack on the surface of optical glass BK7 when scratched by a Vickers indenter [27]. The influence of the load and loading time of the Vickers indenter on the mechanical properties of the glass was studied by J. Guin et al. [28]. The critical loads of the appearance of microcracks during scratching, as well as Young's modulus and micromechanical properties during instrumental indentation on the protective glasses of solar cells, were studied in [29,30]. The study of the indenter geometry influence on crack formation mechanisms was the subject of T. M. Gross' investigation [31]. The results of experimental and theoretical studies of the fracture stages laws and development of cracks in the surface layer of brittle materials depending on the load and external factors by the method of progressive scratching were described in [27,32,33]. At the same time, increased resistance to the damage of the glass subsurface layer, which is not characteristic of subsequent layers of this material, was noticed. A significant amount of research has been devoted to experimental and theoretical studies of the damage stage laws and the growth of cracks in the surface layer of brittle materials depending on the load and external factors by the method of progressive scratching. In this case, the load causing the crack formation was determined directly during scratching.

The influence of the load and the scratching rate on the fracture mechanisms of glass was presented in [34]. The study [35] was devoted to the crack formation particulars during progressive scratching by a spherical indenter. K. Li with colleges studied the effects of water, the indenter shape, and the scratching rate on the glass damage mechanism [36].

The significant variety of factors, such as the environment, scratching speed, indenter shape, load time, and others that affect glass damage during scratching, as well as the large dispersion of experimental results, necessitate an additional study of the damage mechanisms of glasses and crack formation during scratching.

## 3. Experimental Set-up

The glass damage process was studied by depth-sensing indentation (nanoindentation) and microscratching techniques by various types of indenters using multifunctional instrument "Micron-gamma", which is designed for determination of physical and mechanical properties of the material surface layers [37]. The linear translational worktable of the instrument moves with a constant speed of 22 $\mu$m/s. The indenter is loaded by an electromagnet unit with maximum load 4 N. Normal and tangential indenter displacements are measured by inductive sensors with resolution 5 nm.

During translational movement of the work-table with specimen, any loading law can be applied to the indenter with simultaneous registration of its displacement and tangential force (FF) (Figure 1). To measure the surface profile (profilogram), the scanning mode is performed under the small load $F_{Nmin}$ = 1 mN to avoid surface damage (Figure 1a). In the progressive scratch mode, the load applied to the indenter increases up to preset value $F_N$ and then decreased (Figure 1a). In the static scratch mode (Figure 1b), the indenter load during the first second increases from zero up to the preset value $F_N$ and then with constant load continues to scratching the surface of the test specimen for a given length. The dependence of changes in friction force $F_F$ (tangential force, s. Figure 1b)—tribogram and indenter penetration depth under load during scratching—in the sclerogram was continuously recorded during testing.

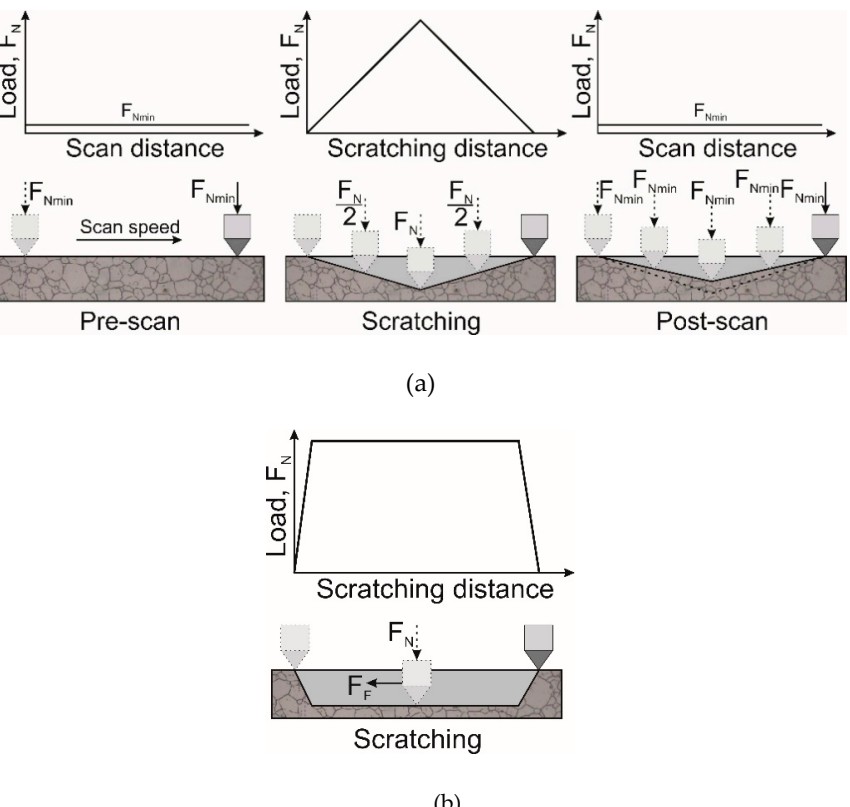

**Figure 1.** Schemes of progressive (**a**) and static (**b**) specimens scratching.

For a detailed analysis morphology of the scratches formed on the tested specimens surfaces the three-dimensional optical profilometer "Micron-alpha" was used (Figure 2). The instrument was designed on the base of Linnik interferometer and allows us to measure surface topography with nanometer axial resolution. The principle of its operation is based on white lite interferometry, an established and proven method for the precise measurement of surface shape [38].

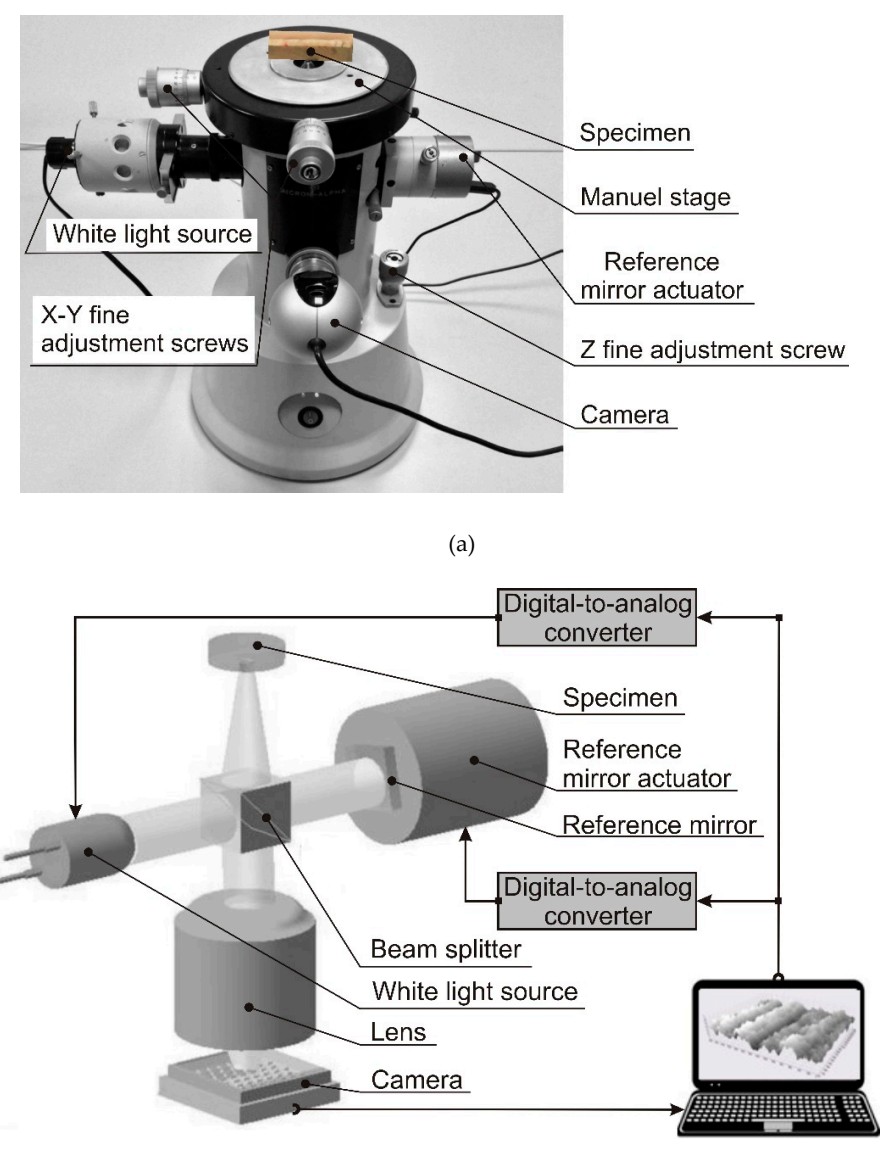

(a)

(b)

**Figure 2.** General view (**a**) and schematic diagram (**b**) of the three-dimensional interference profilometer.

A white light source emits a light to the beam splitter. The brightness of this source is controlled by a computer through digital-to-analog convertor (DAC). It divides the beam into two parts and directs them to sample surface and reference mirror. The light is reflected back to the beam splitter joint and forms interference fringes focused by the lens on the camera. Scanning is achieved by an electromagnetic reference mirror actuator which changes the optical path length of the reference beam. The scanning is controlled by software via 24-bit DAC. During scanning, the light intensity is recorded by camera. The interference signal has a maximum intensity when the optical path difference is equal to zero. A general view of the optical profilometer and its main functional components is shown in Figure 2a. Figure 2b shows the schematic diagram of this profilometer.

Studies were carried out on different float, optical and other glasses (Table 1). The experimental tests were performed on polished specimens with surface roughness parameters $R_a$ = 0.003–0.01 μm and $R_z$ = 0.02–0.05 μm. The specimens have a rectangular bar shape with length 20–40 mm and 3–4 mm cross section (earlier tested by EF [11]). The features of glass failure were studied due to the changes in penetration depth of diamond Berkovich indenter during its edge forward movement. The scratch tests were done in progressive mode. The scratch length was 2 mm (Figure 1a) with maximum normal load on indenter 1200 mN, which is similar to the data according to the accepted world practice of glass failure investigation [25,27].

**Table 1.** Mechanical properties of the studied glasses.

| Material | Microhardness, H (GPa) (Dispersion) | Elastic Modulus, E (GPa) (Dispersion) | Load of Lateral Crack, $F_L$ (mN) (Dispersion) | Fracture Resistance, $F_R$ (N/mm) (Dispersion) | Fracture Toughness, $K_{IC}$ (MPa•m$^{1/2}$) | Penetration Work, $A_g$ (nJ) | Ductility Coefficient, $\delta$ (-) |
|---|---|---|---|---|---|---|---|
| Flint TF-1 | 5.6 (<1%) | 58.7 (<1%) | 700 (45%) | 192 ± 41 | 0.85 | 45.30 | 0.478 |
| Heavy flint F-2 | 5.9 (<1%) | 60.7 (<1%) | 1000 (40%) | 192 ± 31 | 0.83 | 43.78 | 0.473 |
| Heavy crown TK-14 | 8.9 (<1%) | 96.2 (<1%) | 400 (30%) | 171 ± 48 | 0.51 | 35.90 | 0.492 |
| Fused silica KI | 11.3 (<1%) | 71.2 (<1%) | 300 (20%) | 224 ± 66 | 0.77 | 37.07 | 0.326 |
| Float glass | 8.4 (<1%) | 76.9 (<1%) | 600 (30%) | 241 ± 55 | 1.03 | 37.76 | 0.425 |
| Light krone LK-5 | 8.8 (<1%) | 78.1 (<1%) | 400 (25%) | 219 ± 62 | 0.91 | 37.90 | 0.406 |
| Neodymium glass | 7.8(<1%) | 73.1(<1%) | 300 (35%) | 228 ± 50 | 0.84 | 38.87 | 0.441 |
| Quartz glass | 11.1 (<1%) | 69.7 (<1%) | 500 (30%) | 220 ± 66 | 0.80 | 36.46 | 0.311 |
| Soda-lime glass | 6.5 (<1%) | 73.4 (<1%) | 800 (45%) | 241 ± 55 | 0.66 | 41.96 | 0.482 |

Instrumented indentation measurements of glasses were performed using a Berkovich diamond tip. Microhardness, modulus of elasticity and plasticity characteristic were calculated from indentation diagrams with method of Oliver W. C. and Pharr G. [39]. Microphotographs of the scratches and indents were registered using the instrument's build-in microscope.

## 4. Results

Before studying the fracture process of glasses, their mechanical properties were determined using a depth-sensing indentation test at a maximum load of 500 mN with a loading rate of 50 mN/s using "Micron-gamma" instrument. The maximum indentation depth was from 1.85 to 2.3 μm, depending on the glass material. Figure 3 shows nanoindentation diagrams of the glasses test.

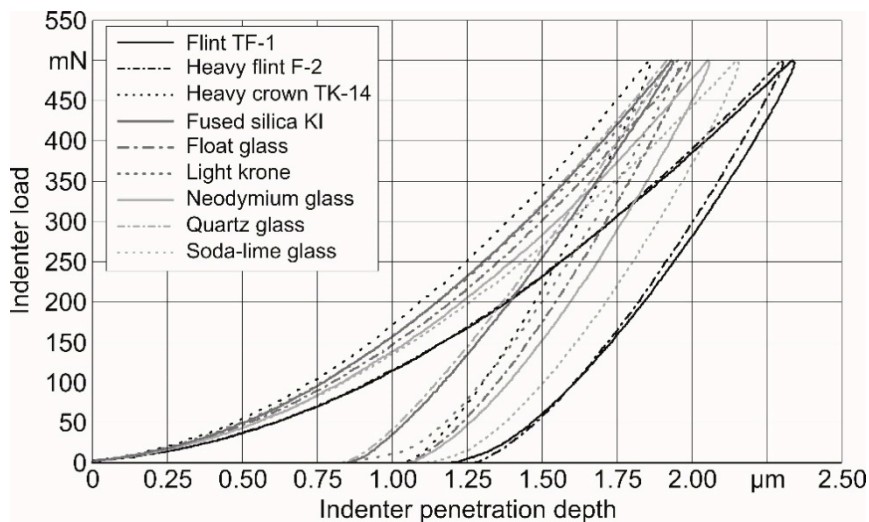

**Figure 3.** Diagrams of the indentation into the studied glasses surface.

Mechanical properties of the studied glasses, determined by instrumented nanoindentation tests, are presented in the Table 1. Additionally, Table 1 shows the previously measured results of

fracture resistance F$_R$ and fracture toughness K$_{1C}$ using Rockwell indenter [10,11]. The plasticity index δ determines a part of the work expended on material plastic deformation during indenter penetration [40]. To understand the load influence on the glass surface failure during scratching, tests were done in progressive mode with a loading rate of 5 mN/s. As an example, tribogram is presented in Figure 4, and Figure 5 shows a sclerogram of the scratch in the glass.

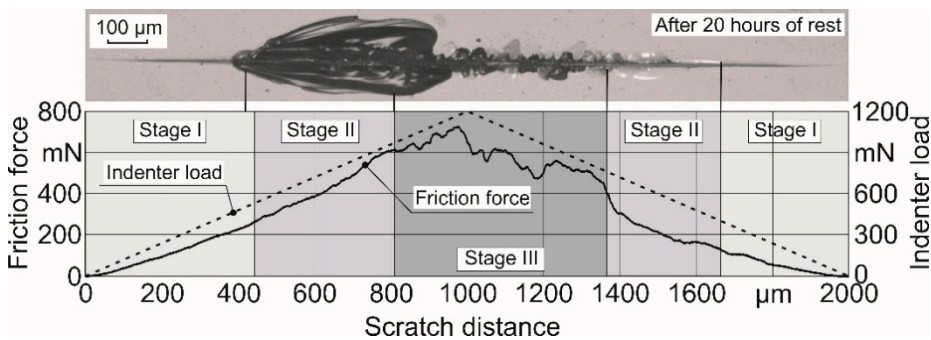

**Figure 4.** Tribogram and micro photo of the scratch in the glass light krone LK-5 by the progressive method.

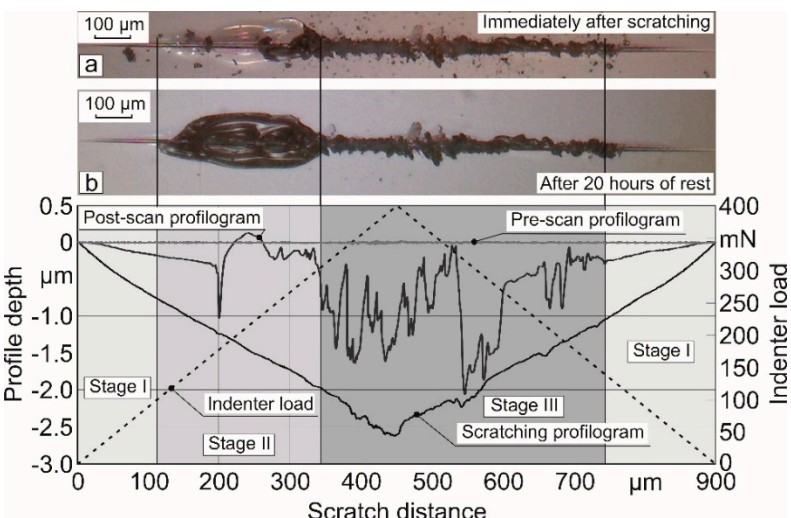

**Figure 5.** Diagrams and scratch micro photos immediately after scratching (**a**) and after 20 h of rest (**b**) in the quartz glass by the progressive method.

Three stages of micro-failure of the glasses test are formed as the load on the indenter increases during the scratch test. Similar stages, however, under different test conditions, were observed in the study [12]. The initial stage of viscous-brittle micro-fracture (Stage I) ends at the scratch length 440 μm, when the load reaches 500 mN (Figure 4). Then, the second stage of lateral crack formation begins (Stage II), as the load increases to 1000 mN in the area from 440 to 820 μm. A further increase in load to 1200 mN, in the area from 820 to 1000 μm, leads to the third (Stage III) stage of fracture. At this stage, the growth of lateral crack stops and radial cracks begin to form, creating a narrow and deep part of the scratch with fragile shores. Moreover, this stage continues despite a decrease in load to 750 mN in the area 1000–1370 μm. Further load decreasing leads to the second stage of lateral crack formation in the area 1370–1660 μm (Stage II) and then to the first stage (Stage I). During load decrease in the area from 1000 to 1600 μm, the third stage, missing the second one, passes into the first. In this case, lateral cracks do not form.

It should be noted the further propagation of lateral cracks formed in the second stage of fracture. These cracks continue propagate after the scratch test for a long-time followed formation of chips (Figure 5). The crack propagation time ranged from a few seconds to many hours (Figures 4 and 5).

Quartz glass was used as a reference material for a closer look at the mechanism of crack initiation and propagation in glasses. Preliminary studies have shown that lateral cracks initiate in this glass at lower loads on the indenter than in such glasses as light krone LK-5, heavy flint F-2 and others. Therefore, a progressive scratch test on the quartz glass was carried out at a maximum load applied to indenter of 400 mN in the center of the scratch. The scratch test includes three steps, which follow the same path over the surface.

In the first step, the surface profile and slope were measured at the load on the indenter of 1 mN (Figure 5, pre-scan). Such a small load makes it possible to measure the initial surface profile without visible damage. Pre-scan is an analogue of profilogram, taking into account the Berkovich indenter tip geometry. Then, returned to the initial position, progressive scratching was performed on the same path with a maximum indenter load of 400 mN in the middle of the scratch and the sclerogram was recorded during testing (Figure 5). At the end of this test, the third step (post-scan) was performed with an indenter load of 1 mN. After the indenter had been returned to the starting point, a post-scan was performed in order to measure scratch profile and the resulting damages (Figure 5, post-scan).

A micro photo of the scratch in the quartz glass immediately after the carrying out of progressive scratch test is shown in Figure 5a. It can be seen on the photo that above the lateral crack, a pill-up of the glass surface area (as indicated by arrow on post-scan profile) relative to the initial surface (pre-scan profile) is observed. A micro photo of the same scratch after 20 h of rest is presented in Figure 5b. The micro photo shows that the chip was formed due to the further propagation of the lateral crack. This chip was probably formed under the influence of residual internal stresses, as well as a result of water vapor adsorption from the air in accordance with the well-known Rehbinder effect.

As can be seen from the sclerogram (Figure 5), the indenter smoothly penetrates into the sample surface and only closer to the middle of the scratch, when the load reaches maximum value, the smoothness (slope) of its penetration changed due to the glass fracture. However, the micro photo (Figure 5a) and post-scan scratch profile indicate that glass fracture was initiated much earlier and at lower indenter loads. This indicates that the fracture zone propagates along the scratch for some time in the direction opposite to the indenter movement (Figure 5b).

Therefore, it is difficult to reliably measure the value of lateral crack initiation load. A lateral crack is initiated under moving indenter and begins to propagate after the load is removed. A further propagation of the lateral crack occurs within the next 20 h after the load is removed.

Similar failure stages during scratching are observed on almost all studied glasses, even obsidian glass. For example, scratch diagrams for obsidian glass are presented in Figure 6. Micro photo and scratch profiles clearly demonstrate three stages of glass fracture: Stage I—viscous-brittle fracture; Stage II—lateral crack initiation; Stage III—radial crack initiation.

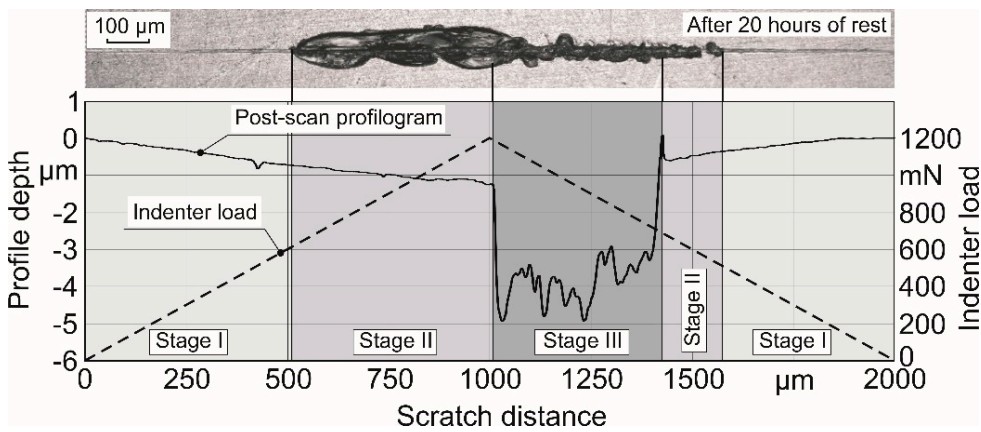

**Figure 6.** Diagrams and micro photo of the scratch in the obsidian glass by the progressive method.

To assess the effect of the time factor on the lateral crack propagation in glass (Figures 5 and 6), scratch tests were performed at a load that did not instantly initiate cracks (Figure 7).

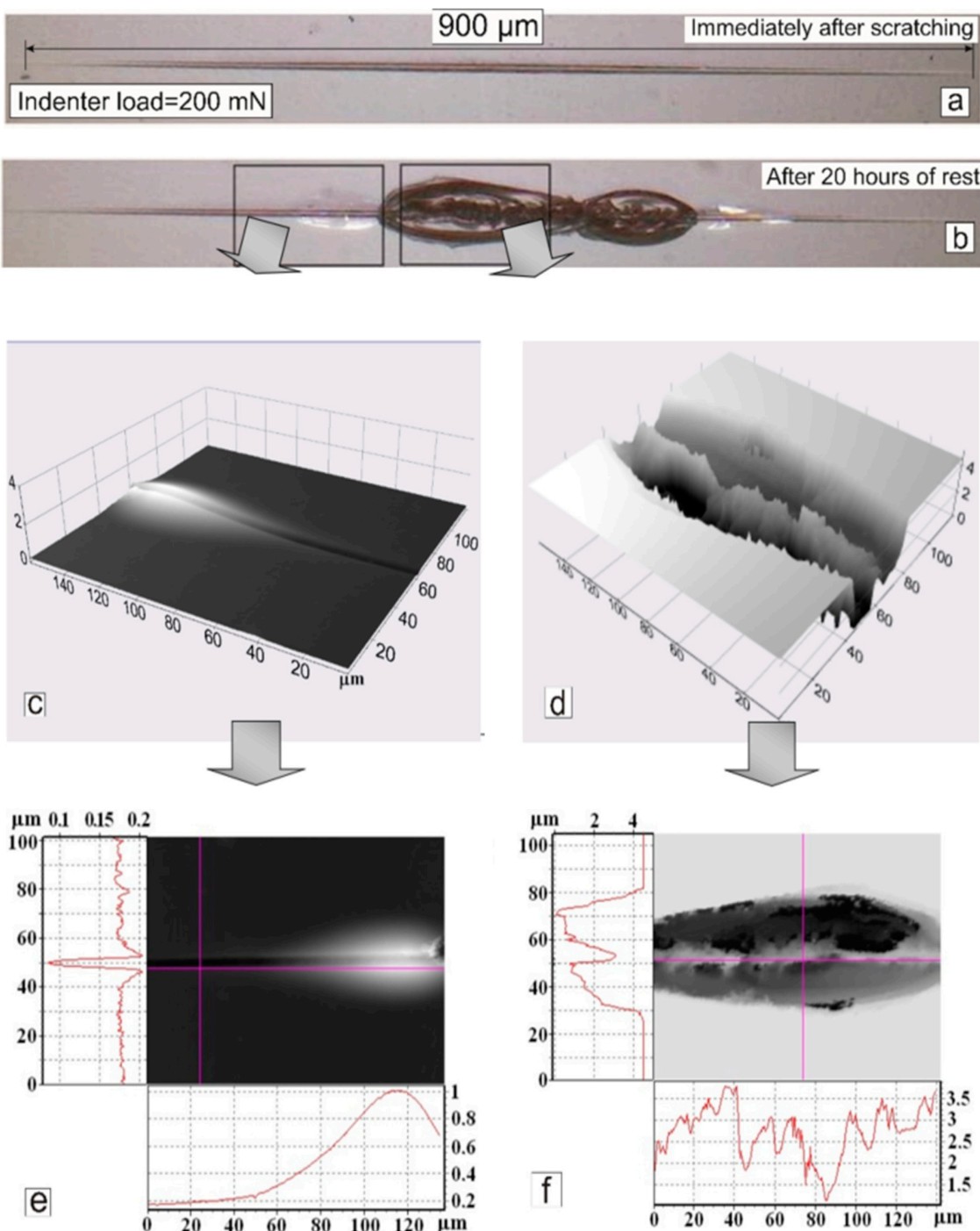

**Figure 7.** Scratch micro photos immediately after scratching (**a**) and after 20 h of rest (**b**) as well as enlarged 3D images (**c**) and (**d**) and 2D images (**e**) and (**f**) of selected areas of the profile at a maximum load of 200 mN.

Therefore, scratching of quartz glass surface using progressive method were performed at a maximum normal load of 200 mN. Such a load is not enough for lateral crack initiation, as can be seen in the micro photo, which was obtained immediately after scratching (Figure 7a). However, the lateral cracks were found along the same scratch 20 h after the experiment. Most of these cracks are turned

into chips (Figure 7b). A part of the lateral crack is formed at a lower load, raising the upper glass layer above the surface by approximately 1 μm (Figure 7c). Another part of the crack throws the material in different directions, forming chips (Figure 7d). Such a fracture mechanism is confirmed by profile analysis of the formed scratch (Figure 7e,f). These chips are located along the scanning path. They have almost vertical shores with a depth of 4–5 μm and consist of two parts, alternating elevation in the middle.

During progressive scratching of quartz glass, three stages of its destruction are also observed (Figures 5 and 7). At Stage I of scratching, the damage is viscous-brittle. The load on the Berkovich indenter was less than 200 mN and the scratch profile was similar to the indenter shape (Figure 7c,e). At Stage II, when the indenter load reaches values from 200 to 300 mN, two symmetrical, with respect to the scanning path, lateral cracks are formed. These cracks can grow over time, turning into oval craters (chips) with widths from 100 to 200 μm and depths from 3 to 4 μm. With an increase in normal load over 300 mN, Stage III of glass failure occurs. At this stage, the brittle fracture of the glass surface layers occurs, caused by the formation of radial cracks under the moving indenter. A narrow scratch with sharp edges and width from 20 to 30 μm that does not change over time is formed (Figure 7d,f).

The fixed fact of lateral crack propagation along the scratch for a long time allows us to make the assumption that the scratch length has an effect on the consistent patterns of such crack initiations. To assess the effect of scratch length on the lateral crack propagation in glass, studies were made of scratch behavior with scientifically shorter length compared to conventional practice (1 mm and longer) [21,23].

Separate scratches with lengths of 270 μm were applied on the surface of the quartz glass specimens with discretely increasing normal load (Figure 1b). Lateral cracks resulting from this were localized within these short scratches. During the scratch test with a constant load pf 150, 250 and 350 mN, sclerograms and scratch profiles (post-scans) were recorded (Figure 8). Additionally, scratch micro photos are shown near the corresponding scratch profile. A lateral crack propagating along a scratch is localized around a short scratch. Therefore, such a discrete loading of the indenter when applying short scratches makes it possible to more accurately determine the magnitude of the lateral crack formation load. Micro photos of scratches in glasses at different normal loads shows three stages of glass fracture: Stage I—viscous-brittle fracture; Stage II—lateral microcrack initiation; and Stage III—radial crack initiation. Moreover, the area of surface damage in Stage II is much larger than the area on Stage III. This indicates a disproportionate increase in the failure area of the scratch with increasing load on the indenter. Apparently, this phenomenon is associated with the presence of a significant gradient of residual stresses in depth.

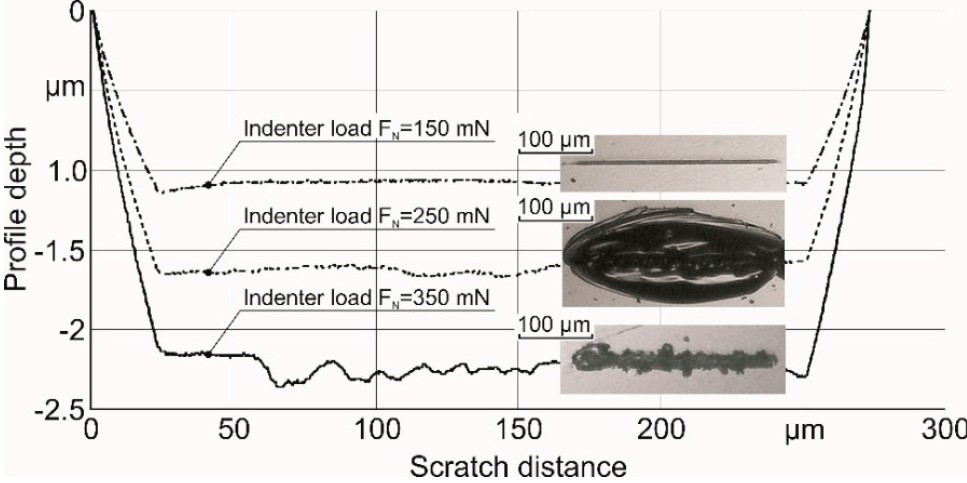

**Figure 8.** Scratching sclerogramm and micro photos of short scratches in the quartz glass.

Similarly, destruction of glasses such as light krone LK-5, float glass, heavy flint F-2, flint TF-1, obsidian, neodymium, F2 and quartz glass was investigated. During tests, lateral crack initiation loads were recorded (Figure 9). Formed lateral cracks in different glasses have a similar look and shape of the chips. The differences are observed only in crack initiation load.

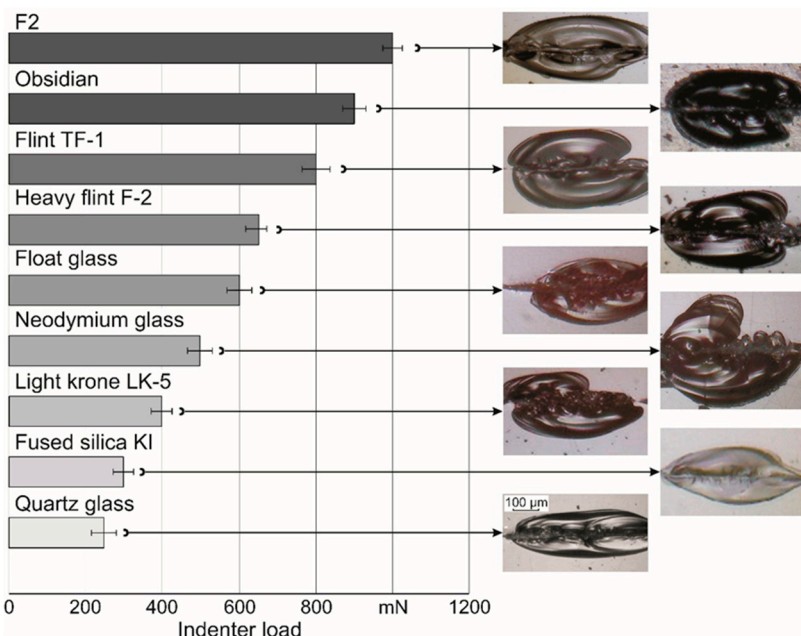

**Figure 9.** Lateral cracks by discrete scratching of glasses.

Two constant load scratches were made on the glass surfaces to verify the assumptions of lateral crack propagation after the tested sample surface was scratched. The first scratch was made on the heavy flint F-2 surface with a constant indenter load of 400 mN. A second scratch was made with the same indenter load, but after a pause of 10 s and at a distance 5 µm from the first one (Figure 10). Thus, the two scratches were separated relative to each other. The distance between the two identical scratches that have the length 270 µm was approximately 5 µm. The time of scratch test procedure for one scratch was 12 s and the indenter movement speed was 22 µm/s. The total time of the test for two scratches was 34 s, respectively. The next pairs of scratches were done with an indenter load of 100 mN, in exactly the same way (Figure 10).

The obtained experimental results (Figure 10) confirm the assumption on the long period of time lateral crack propagation initiated by the scratching. Crack propagation can take several days. Two scratches that were made with insufficient load for lateral crack initiation 400 mN stay separated (Figure 10). Scratches, which were made with the load 500 mN, which was sufficient for lateral crack initiation, were combined into one large chip. In the same way, two separate scratches that were made in the sample at indenter loads of 600 and 700 mN were combined together into one large chip. However, increasing the scratching constant load to more than 700 mN (from 800 to 1200 mN) leads to the radial crack initiation causing brittle fracture under the indenter (Figure 10). This happens because high stresses instantly destroy the specimen surface, bypassing the conditions for the formation of lateral cracks on its surface.

To study water's effect on the glass surface's destruction processes during scratching, additional studies with an indenter load of 1600 mN and scratch length of 150 µm were carried out (Figure 11).

On the float glass surface cleaned by acetone, a scratch was applied by eight repeated passes. Three-dimensional surface topography of the obtained scratch is shown in Figure 11a. Then, a series of repeated scratches were made in the drop of water at a distance 300 µm. Brittle fracture of glass in the water was observed after the second pass (Figure 11b).

The obtained results confirm the significant influence of water on the glass destruction during scratching. Significant glass surface fracturing in water occurred after the second indenter pass, whereas after eight repeated indenter passes in the absence of water, a straight scratch without visible brittle fracturing of its shores was formed. Water contributes to glass surface destruction, penetrates into the micro cracks formed during scratching, and has a wedging effect.

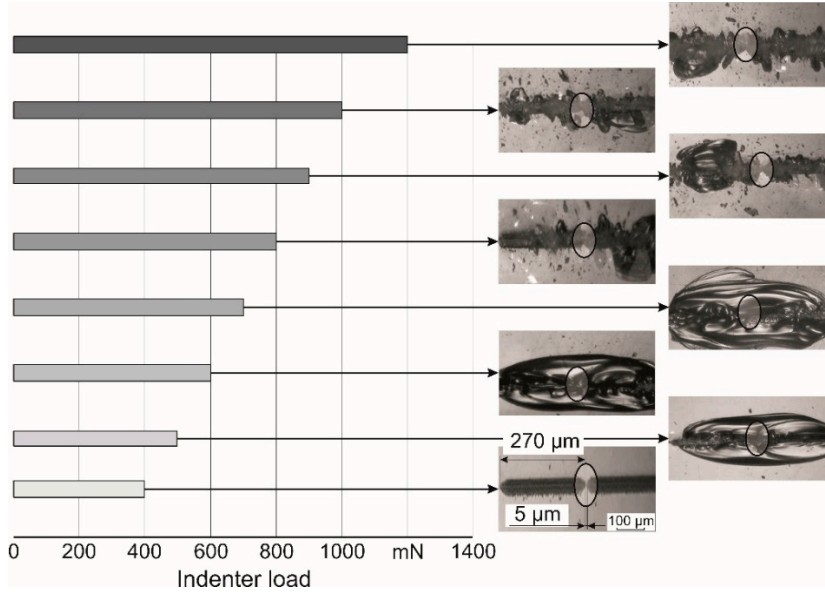

**Figure 10.** Micro photos of two adjacent scratches 270 μm long, obtained by scratching heavy flint TF-2 glass with different indenter loads.

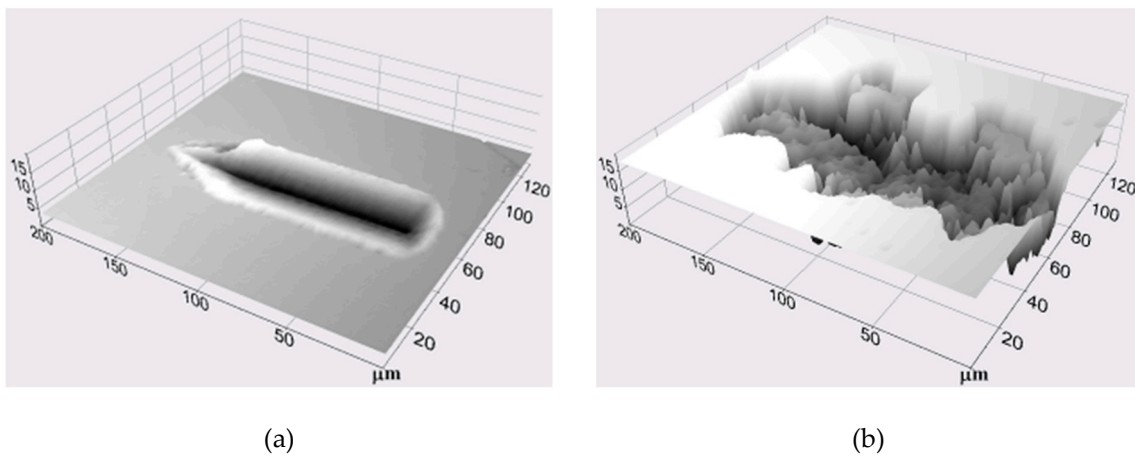

(a)                                                                    (b)

**Figure 11.** Scratch profile on the float glass surface in the air (**a**) and in the water (**b**).

## 5. Discussion

It can be assumed that the stages of lateral chip formation are strictly sequential and they are formed not only during the scratching process, but also after the sample surface was scratched. Lateral cracks could propagate along thin scratch shores over time and then are turned into large chips. Lateral crack initiation and propagation contributed by the stresses are formed during scratching, exacerbated by adsorbed water vapors from the air. Surface-active water molecules penetrate into the micro cracks and push them apart, due to the wedging action better known as the Rehbinder effect [41]. This contributes further to crack propagation.

Based on the experimental studies' results, a model is proposed for the glass destruction during scratching by the Berkovich indenter (Figure 12).

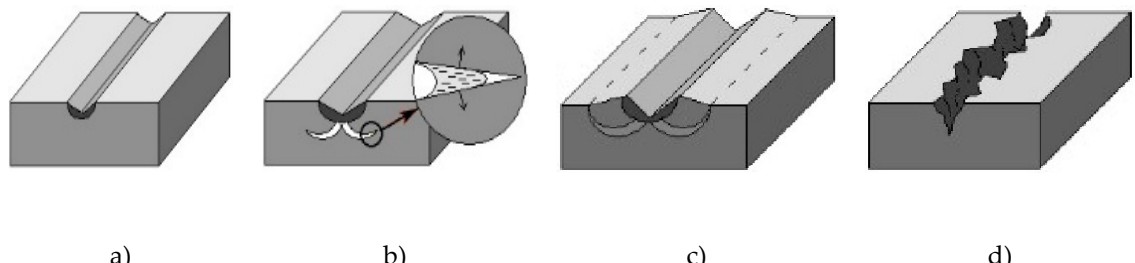

a)        b)        c)        d)

**Figure 12.** Glass breakage model when scratched: Stage I (**a**), Stage II (**b**), grow of cracks (**c**), formation of brittle fracture and sharp edges (**d**).

Glass failure goes through three stages during load increasing. At Stage I, with small indenter loads, viscous-brittle fracturing occurs, which forms a scratch similar to the indenter tip shape (Figure 12a). In Stage II, as the indenter load increases, lateral cracks are initiated at the depth from 2 to 5 μm (Figure 12b). These cracks grow over time under the influence of residual stresses and the wedging effect of absorbed water vapor from the air (Figure 12c). Cracks are turned into wide chips with a width from 150 to 300 μm. At Stage III, with a further increase in indenter load to the critical value, the mechanism of crack formation changes. Radial cracks appear and brittle chipping of glass fragments occurs directly under the indenter during the scratching process. Scratches with a brittle fracture and sharp edges are formed (Figure 12d).

## 6. Conclusions

The experimental studies of glass microfracture have shown that with the progressive scratch method, lateral cracks propagate further along the scratch after it has been applied. Lateral cracks continue to develop after the end of the scratching process for a long time, up to many hours. This makes it difficult to determine the value of the load causing the crack.

A method for scratching the surface of the test glass under constant load and on short traces has been proposed and tested. This provides crack localization within a short scratch. As a result, this allows for determining the load of lateral cracking.

During the experiments, a significant effect of the medium on the process of glass destruction was found. For example, in the presence of water, the initiation and propagation of cracks in the surface layers of glass specimens is significantly facilitated. Their brittle fracture occurs with significantly less load on the scratching indenter. The large dispersion of the loads at which cracks initiate is explained by the influence of environmental humidity on glass damage during scratching.

The authors plan to devote the following set of studies to the influence of the medium on the process of glass failure.

**Author Contributions:** Conceptualization, I.Z.; Methodology, I.Z.; Software, M.S. and V.Z.; Validation, V.Z.; Formal analysis, M.S. and I.Z.; Investigation, I.Z. and V.Z.; Resources, G.A.G. and V.Z.; Data curation, M.S., G.A.G. and V.Z.; Writing—original draft preparation, I.Z., M.S. and V.Z.; Writing—review and editing, M.S.; Visualization, I.Z. and M.S.; Supervision, G.A.G.; Project administration, G.A.G.; Funding acquisition, G.A.G. All authors have read and agreed to the published version of the manuscript.

**Funding:** The present results were gained in the project "Study of mechanical behavior of glasses and ceramics at their micro- and macro-destruction" (No. 1.3.4.1112), funded by the National Academy of Sciences of Ukraine (Grant Agreement No. 0111U000373).

**Acknowledgments:** The authors would like to thank the National Academy of Sciences of Ukraine for the support, which is highly appreciated.

**Conflicts of Interest:** The authors declare no conflict of interest.

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
