# Peer review of "Glass Fracture during Micro-Scratching"

_surfaces, doi:10.3390/surfaces3020016_

Round 1
Reviewer 1 Report
It is an extensive study on glass fracture induced by microscratching. The methods used are appropriate and well explained. A large number of results obtained and well justified.
The only comment that I have is a stange unit annotation on graphs, they are inserted between numbers rather than next to the axis description.
Author Response
The authors are very grateful to Reviewer 1 for meticulously reviewing and interpreting the content of the paper.
Reviewer #1: It is an extensive study on glass fracture induced by microscratching. The methods used are appropriate and well explained. A large number of results obtained and well justified.
“The only comment that I have is a stange unit annotation on graphs, they are inserted between numbers rather than next to the axis description.”
- Entering the dimension of the values to be displayed instead of the penultimate number of axes in diagrams has been practiced in our publications for decades. For example, you could look at our published articles:
- Storchak, M.; Rupp, P.; Möhring, H.-C.; Stehle, T.: Determination of Johnson–Cook Constitutive Parameters for Cutting Simulations. Metals, 2019, Vol. 9, 473.
- Möhring, H. C., Kushner, V., Storchak, M., & Stehle, T.: Temperature calculation in cutting zones. CIRP Annals - Manufacturing Technology, 2018, Vol. 67, pp. 61 – 64.
- Storchak, M., Zakiev, I., & Träris, L.: Mechanical Properties of Subsurface Layers in the Machining of Titanium Alloy Ti10V2Fe3Al. Journal of Mechanical Science and Technology, 2018, Vol. 32, No. 1, pp. 315 – 322.
and several others.
However, if the reviewer insists on the location of the dimension in the name of the axis, then the authors will modify all diagrams in accordance with the wishes of the reviewer.
Reviewer 2 Report
The research work is interesting. The reviewer wants to know about the following comments:
1. On page 2, lines 91-92: K. Li with colleges studied the water, the indenter shape, and the scratching rate effects on the glass damage mechanism [28], here colleges stand for.
2. The reviewer wants to see recent more literature of glass damage mechanism.
3. On page 5, line 161: „Micron-gamma“ instrument (s. Chapter 3). The reference is needed regarding (s. Chapter 3) and it looks good if „Micron-gamma“changes into "Micron-gamma“.
4. On page 7, lines 229-230: three stages of glass fracture: Stage I – viscous-brittle fracture, Stage II – lateral crack initiation, Stage III – radial crack initiation were mentioned in the manuscript. Did the authors find such three stages of glass fracture in the literature? If so, references are needed.
5. On page 9, lines 277-278: tree stages of glass fracture: Stage I – viscous-brittle fracture, Stage II – lateral microcrack initiation и Stage III – radial crack initiation. Here, the reviewer wants to know the meaning of tree stages and и Stage III.
6. On page 10, lines 315-317: This happens because high stresses instantly destroy the specimen surface, bypassing the conditions for the formation of lateral cracks on its surface. Did the authors find such conditions in the literature?
7. Elaborate discussion by comparing the results with the literature.
8. On page 2, lines 58-62: detect plagiarism.
9. The English language is necessary to correct throughout the manuscript.
Author Response
The authors are very grateful to Reviewer # 2 for meticulously reviewing and interpreting the content of the paper.
Reviewer #2: The research work is interesting. The reviewer wants to know about the following comments:
- “On page 2, lines 91-92: K. Li with colleges studied the water, the indenter shape, and the scratching rate effects on the glass damage mechanism [28], here colleges stand for.”
- The authors do not understand this comment. We kindly ask the reviewer to spread this comment.
- “The reviewer wants to see recent more literature of glass damage mechanism.”
- The literature sources are inserted, according to the reviewer's comment.
- “On page 5, line 161: „Micron-gamma“ instrument (s. Chapter 3). The reference is needed regarding (s. Chapter 3) and it looks good if „Micron-gamma“changes into "Micron-gamma“.?”
- The authors do not understand this comment. We kindly ask the reviewer to spread this comment.
- “On page 7, lines 229-230: three stages of glass fracture: Stage I – viscous-brittle fracture, Stage II – lateral crack initiation, Stage III – radial crack initiation were mentioned in the manuscript. Did the authors find such three stages of glass fracture in the literature? If so, references are needed.”
- The literature source is inserted, according to the reviewer's comment.
- “On page 9, lines 277-278: tree stages of glass fracture: Stage I – viscous-brittle fracture, Stage II – lateral microcrack initiation и Stage III – radial crack initiation. Here, the reviewer wants to know the meaning of tree stages and и Stage III.?”
- The meaning of these stages was in the Text of manuscript (p. 9, lines 277-279): “Stage I – viscous-brittle fracture, Stage II – lateral microcrack initiation и Stage III – radial crack initiation”. Therefore, the authors do not understand this comment. We kindly ask the reviewer to spread this comment.
- “On page 10, lines 315-317: This happens because high stresses instantly destroy the specimen surface, bypassing the conditions for the formation of lateral cracks on its surface. Did the authors find such conditions in the literature?”
- The authors found the study ([12]) of lateral cracks formation by scratching of glass. The corresponding literature is inserted (s. answer 4).
- “Elaborate discussion by comparing the results with the literature.”
- A new model of damage of glass is proposed in the discussion. No similar models exist in the literature. Therefore, the chapter "Discussion" cannot be compared with the known literature sources.
- “On page 2, lines 58-62: detect plagiarism.”
- On page 2, lines 58-62 analyze the authors well-known literature sources by entering appropriate references. The authors do not understand where does the reviewer see the plagiarism. Therefore, we kindly ask the reviewer to explain this claim.
- “The English language is necessary to correct throughout the manuscript.”
- The authors have checked the entire text of the manuscript for language consistency. Some corrections have been made in the text. However, it would be very helpful if the reviewer entered two or three examples of language errors. This served as a very big help for the author.
Round 2
Reviewer 2 Report
The reviewer wants to give thanks to the authors for improving the manuscript and would like to know about the following comments:
- “On page 2, lines 91-92: K. Li with colleges studied the water, the indenter shape, and the scratching rate effects on the glass damage mechanism [28], here colleges stand for.” I think instead of colleges, I suggest colleagues.
- “On page 5, line 161: „Micron-gamma“ instrument (s. Chapter 3). The reference is needed regarding (s. Chapter 3) and it looks good if „Micron-gamma“changes into "Micron-gamma“.?” I mean the authors need to put inside the inverted commas as "Micron-gamma".
- “On page 9, lines 277-278: tree stages of glass fracture: Stage I – viscous-brittle fracture, Stage II – lateral microcrack initiation и Stage III – radial crack initiation. Here, the reviewer wants to know the meaning of tree stages and и Stage III.?” I have noticed the strange symbol of " и".
- “On page 2, lines 58-62: detect plagiarism.” The authors need to write their own languages even in their own cited paper. The reviewer checked plagiarism with the help of (http://www.ithenticate.com/)
- “The English language is necessary to correct throughout the manuscript.” Some examples of the English problem are as comments 1, 2, and 3.
Author Response
The authors are very grateful to Reviewer # 2 for interpreting and explaining the comments.
Reviewer #2: The reviewer wants to give thanks to the authors for improving the manuscript and would like to know about the following comments:
- “On page 2, lines 91-92: K. Li with colleges studied the water, the indenter shape, and the scratching rate effects on the glass damage mechanism [28], here colleges stand for.” I think instead of colleges, I suggest colleagues.”
- The text is corrected according to the reviewer's comment.
- “On page 5, line 161: „Micron-gamma“ instrument (s. Chapter 3). The reference is needed regarding (s. Chapter 3) and it looks good if „Micron-gamma“changes into "Micron-gamma“.?” I mean the authors need to put inside the inverted commas as "Micron-gamma”
- The commas are corrected, according to the reviewer's comment.
- “On page 9, lines 277-278: tree stages of glass fracture: Stage I – viscous-brittle fracture, Stage II – lateral microcrack initiation и Stage III – radial crack initiation. Here, the reviewer wants to know the meaning of tree stages and и Stage III.?” I have noticed the strange symbol of " и”
- The text is corrected, according to the reviewer's comment.
- “On page 2, lines 58-62: detect plagiarism.” The authors need to write their own languages even in their own cited paper. The reviewer checked plagiarism with the help of (http://www.ithenticate.com/)”
- The text is changed, according to the reviewer's comment.
- “The English language is necessary to correct throughout the manuscript.” Some examples of the English problem are as comments 1, 2, and 3.”
- The authors have checked again the entire text of the manuscript for language consistency.